# Proteome and Interactome Linked to Metabolism, Genetic Information Processing, and Abiotic Stress in Gametophytes of Two Woodferns

**DOI:** 10.3390/ijms241512429

**Published:** 2023-08-04

**Authors:** Sara Ojosnegros, José Manuel Alvarez, Jonas Grossmann, Valeria Gagliardini, Luis G. Quintanilla, Ueli Grossniklaus, Helena Fernández

**Affiliations:** 1Area of Plant Physiology, Department of Organisms and Systems Biology, University of Oviedo, 33071 Oviedo, Spain; uo286037@uniovi.es (S.O.); alvarezmanuel@uniovi.es (J.M.A.); 2Functional Genomic Center Zurich, University and ETH Zurich, 8092 Zurich, Switzerland; jg@fgcz.ethz.ch; 3Swiss Institute of Bioinformatics, 1015 Lausanne, Switzerland; 4Department of Plant and Microbial Biology & Zurich-Basel Plant Science Center, University of Zurich, 8008 Zurich, Switzerland; vgagliar@botinst.uzh.ch (V.G.); grossnik@botinst.uzh.ch (U.G.); 5Department of Biology and Geology, Physics and Inorganic Chemistry, University Rey Juan Carlos, 28933 Móstoles, Spain; luis.quintanilla@urjc.es

**Keywords:** *Dryopteris affinis* ssp. *affinis*, *Dryopteris oreades*, fern, gametophyte, non-seed plant, proteome, STRING database

## Abstract

Ferns and lycophytes have received scant molecular attention in comparison to angiosperms. The advent of high-throughput technologies allowed an advance towards a greater knowledge of their elusive genomes. In this work, proteomic analyses of heart-shaped gametophytes of two ferns were performed: the apomictic *Dryopteris affinis* ssp. *affinis* and its sexual relative *Dryopteris oreades*. In total, a set of 218 proteins shared by these two gametophytes were analyzed using the STRING database, and their proteome associated with metabolism, genetic information processing, and responses to abiotic stress is discussed. Specifically, we report proteins involved in the metabolism of carbohydrates, lipids, and nucleotides, the biosynthesis of amino acids and secondary compounds, energy, oxide-reduction, transcription, translation, protein folding, sorting and degradation, and responses to abiotic stresses. The interactome of this set of proteins represents a total network composed of 218 nodes and 1792 interactions, obtained mostly from databases and text mining. The interactions among the identified proteins of the ferns *D. affinis* and *D. oreades*, together with the description of their biological functions, might contribute to a better understanding of the function and development of ferns as well as fill knowledge gaps in plant evolution.

## 1. Introduction

Ferns and lycophytes represent a genetic legacy of great value, being descendants of the first plants that evolved vascular tissues about 470 million years ago. They are distributed throughout the world and play an important role in ecosystem functioning. Compared to angiosperms, they have received scant attention, relegating them to the background after a splendid past. The aesthetic appeal of their leaves and their use to alleviate ailments in traditional medicine is all that these plant groups have traditionally inspired. Specifically, fern gametophytes are ideal organisms for research on plant growth and reproduction, which is facilitated by a simple in vitro culture system and their small size of a few millimeters. In relation to climate change and other environmental events, ferns can also provide insights into adaptation, for example, they survived periods of high CO_2_ levels. Only a handful of species have been used to delve into basic developmental processes, such as photomorphogenesis [1], spore germination [2,3,4], cell polarity [5], cell wall composition [6], or reproduction. These studies focused on the gametophyte generation, constituted by an autonomously growing organism, which is well-suited for in vitro culture and sample collection [7,8]. Although fern gametophytes possess a very simple structure consisting mostly of a one-cell-thick layer, they display some degree of complexity: apical-basal polarity, dorsoventral asymmetry, rhizoids, meristems in the apical or lateral parts, reproductive organs (male antheridia and female archegonia), and trichomes distributed over the entire surface.

From a metabolic point of view, ferns and lycophytes contain many secondary metabolites, such as flavonoids, alkaloids, phenols, steroids, etc., and exhibit various bioactivities, including antibacterial, antidiabetic, anticancer, antioxidant, etc. [9]. The therapeutic use of both plant groups is changing from its use in the traditional medicine of different peoples to current applications, in which these plants are used to generate nanoparticles [10]. Finally, the use of ferns and lycophytes was recently advocated to address problems caused by biotic and abiotic stresses. Drought is one of the most severe abiotic stresses affecting plant growth and productivity, and ferns and lycophytes could contribute to better understanding and managing it [11]. Other important adaptations of ferns to extreme environments, such as salinity, heavy metals, epiphytes, or a low invasion of their habitats, were summarized by Rathinasabapathi [12]. Likewise, Dhir [13] highlights the high efficiency of many species of aquatic and terrestrial ferns in extracting various organic and inorganic pollutants from the environment. Recently, researchers have become more interested in these plants, which has been made possible by the advent of high-throughput technologies, such as transcriptomics, proteomics, and metabolomics. In fact, performing molecular analyses in ferns has been elusive, as they exhibit higher chromosome numbers and larger genomes than mosses and seed plants [14], which made it difficult to obtain genomic data. Gene expression, induced by either environmental or developmental conditions, can now be examined in non-model organisms because the required techniques have become more affordable as automation and efficiency have reduced costs.

Some transcriptomic and proteomic datasets have been published for ferns during the last decade. First, physiological and proteome analyses were reported with regard to the mechanism of drought tolerance in the resurrection lycophyte *Selaginella tamariscina* [11]. In 2011, the transcriptome of *Pteridium aquilinum* gametophytes was characterized de novo by pyrosequencing, representing the first complete analysis of the transcriptome of a fern [15]. In the same year, a proteomic analysis of the roots of *Pteris vittata* under arsenic stress was carried out [16]. In *Blechnum spicant,* proteomic profiles of male and female gametophytes were reported, revealing an increase in the amount of defense and stress proteins and a decrease in protein synthesis and photosynthesis when inducing male gametophyte development with antheridiogen pheromones [17]. In 2014, a de novo transcriptome assembly of *Lygodium japonicum* was carried out, and with this information a public database was created: Ljtrans DB [18]. Likewise, proteomic and cytological studies associated with germination and growth of rhizoid tips in the fern *Osmunda cinnamomea* were conducted [4]. In the tree fern *Cyathea delgadii*, a proteomic analysis on stipe explants revealed differentially expressed proteins associated with direct somatic embryogenesis [19]. Overlapping patterns of gene expression in gametophytes and sporophytes of the species *Polypodium amorphum* were analyzed by Sigel and colleagues [20], and in the Himalayan fern *Diplazium maximum*, proteomic analysis revealed multiple adaptive response mechanisms to cope with abiotic stresses [21]. Concerning reproduction, sexual versus apomictic expression profiles were analyzed in *Ceratopteris thalictroides* [22], as well as in the species *Adiantum reniforme* var. *sinense* [23]. In addition, some molecular studies were performed on the fern *Ceratopteris richardii* [5,24]; in one of them, sex determination was found to be accompanied by changes in the transcriptome driving epigenetic reprogramming of the young gametophyte [25]; in another, the *WUSCHEL-related homeobox* (*WOX*) gene, which promotes cell division in gametophytes and organ development in sporophytes, was characterized for the first time in ferns [26]; and finally, transcriptional analysis of the young sporophyte showed the conservation of stem cell factors in the root apical meristem [27]. More recently, other studies on ferns were carried out. In 2022, the first complete transcriptome of multiple organs of the fern *Alsophila spinulosa* was obtained, highlighting genes resistant to light stress [28]; its genome was also assembled and characterized; stem anatomy and lignin biosynthesis were investigated [29]; and transcriptomes of several fern species were compared in order to study the adaptive evolution of leaf and root morphology [30]. With the ferns *Dryopteris affinis* ssp. *affinis* [31,32,33] and *Dryopteris oreades* [33,34], both transcriptomic and proteomic analyses were performed by RNA-sequencing and shotgun proteomics using tandem mass spectrometry.

The current work expands our knowledge of proteomic data in ferns, which are far less explored than in seed plants. We present a continuation of previous work [21] on *D. affinis* spp. *affinis* (referred to as *D. affinis* hereafter) and its relative *D. oreades*. We have chosen these species as study models because they represent the two types of reproduction present in fern gametophytes: sexual in *D. oreades* and apomictic in *D. affinis*. Proteins of heart-shaped gametophytes were extracted and identified using a species-specific transcriptome database established in a previous project [31,32,33]. The functional annotation was inferred by blasting identified full-length protein sequences. We report the categorization of proteins that are shared by both sexual and apomictic gametophytes. Specifically, our analysis reveals new proteomic information involved in the metabolism of carbohydrates and lipids, the biosynthesis of amino acids, the metabolism of nucleotides and energy, as well as of secondary compounds, such as flavonoids, terpenoids, lignans, etc., which are important in the plant’s defense against stress. In addition, proteins related to transcription, translation, as well as protein folding, sorting, transport, and degradation are reported.

## 2. Results

A set of 218 proteins shared by the gametophytes of the apomictic fern *D. affinis* (DA) and its sexual relative *D. oreades* (DO) were analyzed using the software programs STRING version 11.5 and CYTOSCAPE version 3.9.1. Proteomic data are available online: https://www.frontiersin.org/articles/10.3389/fpls.2021.718932/full#supplementary-material accessed on 23 June 2023). Therefore, the present work completes previous studies in which 206 and 166 proteins, upregulated in the apomictic and sexual gametophytes, respectively, were analyzed by means of bioinformatic tools, as well as 145 proteins of the remaining 417 that were present in equal amounts in both apomictic and sexual gametophytes. In the present work, the biological functions of the list of 218 proteins were analyzed by using informatic support such as Gene Ontology (GO) and the Kyoto Encyclopedia of Genes and Genomes (KEGG) classifications provided by the STRING platform (Figure 1 and Figure 2). Based on GO classification, most of the identified proteins are involved in known biological functions. In summary, the percentage of proteins associated with the following biological functions are as follows: metabolism of carbohydrates (11.5%), lipids (1.8%), nucleotides (2.2%), biosynthesis of amino acids (6.8%), energy (15.1%), genetic information processing (26.6%), protein folding, sorting and degradation (26.6%), nitrogen and sulfur metabolism (1.8%), and secondary metabolism, including proteins coping with abiotic stress (15.1%).

Some relevant proteins found that will be discussed below are: GLYCINE-RICH RNA-BINDING PROTEIN 3 (RBG3), classified in the group transcription and translation, which functions in RNA processing during stress; FERREDOXIN-NITRITE REDUCTASE 1 (NIR1), grouped in sulfur and nitrogen metabolism, the main activities of which are the reduction of nitrite to ammonium and the improvement of plant assimilation of NO_2_; the protein ENOYL-[ACYL-CARRIER-PROTEIN] REDUCTASE (MOD1), classified in the group metabolism of lipids, involved in fatty acid synthesis and plant morphology; CHALCONE-FLAVANONE ISOMERASE 1 (CHI1), related to the group metabolism of secondary compounds and involved in the flavonoid synthesis; and PROTEIN TRANSLOCASE SUBUNIT SECA1 (SECA1), classified in the group transport, active in protein transfer across thylakoid membrane and plant acclimation. In addition, we found the following proteins in the gametophytes: THYLAKOID LUMENAL 16.5 kDa, grouped in the metabolism of energy and involved in photosynthesis; PHOTOSYNTHETIC NDH SUBUNIT OF LUMENAL LOCATION 5 IC (PNSL5), a protein also classified in the group metabolism of energy that binds to the promoter of the gene *FLOWERING LOCUS D* and represses its expression; ELONGATION FACTOR 2 (LOS1), related to the group transcription and translation and involved in the response to cold; and the protein LEUCINE AMINOPEPTIDASE 1 (LAP1), a chaperone protecting proteins from heat-induced damage.

KEGG classification also revealed that common proteins are mostly associated with the biosynthesis of secondary metabolites, the ribosome, the biosynthesis of amino acids, and protein degradation. These processes include the building of cellular organelles, such as ribosomes or proteasomes (Table 1). Related to ribosomes, there were several protein classes, such as nucleic acid-binding proteins, ribosomal proteins, translation elongation factors, etc. On the other hand, proteasomes mediate the degradation of proteins, and we found proteins of the 20S particle, the proteolytic core, but also regulatory factors.

Protein domains that are abundant in the gametophytes of both ferns were the pyruvate dehydrogenase E1 component and the histidine and lysine active sites of the phosphoenolpyruvate carboxylase, which is involved in carbohydrate metabolism. Regarding the biosynthesis of amino acids, the most abundant domains were aspartate aminotransferase and pyridoxal phosphate-dependent transferase. Among proteins involved in the metabolism of energy, the HAS barrel domain and the F1 complex of the alpha and beta subunits of ATP synthase were abundant. Related to the metabolism of secondary compounds, aromatic amino acid lyase, phenylalanine ammonia-lyase, and the N-terminus of histidase were enriched. Finally, the frequently found domains in proteins involved in transcription and translation were the GTP-binding domain and domain 2 of elongation factor Tu, as well as conserved sites of the ribosomal proteins S10 and S4.

The interactome of the proteins common to *D. affinis* and *D. oreades* represented a network composed of 218 nodes and 1792 interactions (*p*-value < 0.0001). The proteins with the highest number of interactions among the identified proteins are shown in Table 2: LARGE RIBOSOMAL SUBUNIT PROTEIN UL4Z (RPL4A), SMALL RIBOSOMAL SUBUNIT PROTEIN US11X (RPS14C), and SMALL RIBOSOMAL SUBUNIT PROTEIN US17Y (RPS11B) each have 44 interactions.

The strength of the interactions can be weak or strong (Appendix A), using a scale from 0 to 1 where a weak interaction will have a score close to 0 and a strong one a score close to 1. Taking only interactions with a score equal to or greater than 0.99 for each group of proteins studied into account, proteins involved in transcription and translation are those with the highest number of interactions (554), followed by proteins involved in energy (29), carbohydrate metabolism (16), biosynthesis of amino acids (5), and transport (3). According to the STRING software v.11, the evidence of interactions between proteins can be of various types: (a) Experiments: these refer to proteins that have been shown to have chemical, physical, or genetic interaction in laboratory experiments. (b) Databases: this describes interactions of proteins found in the same databases. (c) Text mining: the proteins are mentioned in the same PubMed abstract or the same article of an internal selection of the STRING software v.11. (d) Co-expression: this indicates that the expression patterns of the two proteins are similar. (e) Neighborhood: the genes encoding the proteins are close to each other in the genome. (f) Gene fusion: this indicates that at least in one organism the orthologous genes encoding the two proteins are fused into a single gene. (g) Co-occurrence: this refers to proteins that have a similar phylogenetic distribution. The interactome presented here uses the species *Arabidopsis thaliana* because all the proteins discussed later were found to be homologs of proteins in this species. Therefore, the protein–protein interactions are those expected to be found in *A. thaliana*. On the other hand, STRING collects information from several sources and proposes protein–protein interactions according to the deposited data. We chose to explore all the types of protein–protein interactions published in the STRING platform for our selected proteins.

Next, we consider some of these types of interaction between proteins (Figure 3: text mining, experiments, co-expression, and databases). Specifically, we focused on the group metabolism of carbohydrates (Figure 3a), metabolism of energy (Figure 3b), ribogenesis (Figure 3c), and protein degradation (Figure 3d). Paying attention only to the two main types of evidence for each of these groups, their relationships were analyzed (Figure 4). Evidence from databases and text mining were the most relevant for the metabolism of carbohydrates (Figure 4a), biosynthesis of amino acids (Figure 4b), the metabolism of secondary compounds, and transport, as well as text mining and co-expression data for the metabolism of energy (Figure 4c), and experiments and co-expression data for transcription and translation (Figure 4d). The significance of associations between variables were as follows: in the metabolism of carbohydrates and in the transcription and translation, highly significant in both (*p*-value < 0.001); in the biosynthesis of amino acids, not significant (*p*-value > 0.05); and in the metabolism of energy, marginally significant (*p*-value slightly greater than 0.05).

Alternatively, when comparing the same type of evidence among the different groups of proteins, we observed that the neighborhood interaction was the most important for the biosynthesis of amino acids and transcription and translation; gene fusion for the metabolism of carbohydrates and biosynthesis of amino acids; co-occurrence for the biosynthesis of amino acids and metabolism of secondary compounds; co-expression for the metabolism of energy and transcription and translation; experiments for transcription and translation and transport; evidence from databases for the metabolism of carbohydrates and transport; and, finally, text mining for the metabolism of secondary compounds and transport.

## 3. Discussion

Molecular research conducted in non-model species, such as ferns and lycophytes, is still scarce. These plant groups are the last major lineage of land plants without a reference genome. Their higher chromosome numbers and larger genomes compared to those of mosses and seed plants have contributed to the paucity of reports dealing with genomic or proteomic analyses. Much more effort is needed to broaden the number of species analyzed to complete our knowledge of plant development. Thus, the current work provides novel information on the proteome shared by gametophytes of the apomictic fern *D. affinis* and its sexual relative *D. oreades*, and extends and completes previous studies in these species [31,32,33,34].

Next, we discuss the biological functions and interactions between the set of proteins obtained, which are grouped into three major categories: metabolism, genetic information processing, and response to abiotic stress (Table 3).

### 3.1. Metabolism

Primary and secondary metabolism encompasses a great number of enzymes connecting all the chemical pathways associated with them. Therefore, proteomic analyses usually yield a lot of proteins linked to the biosynthesis or degradation of carbohydrates, lipids, proteins, and nucleotides, as well as others that, albeit being called secondary, are not less important. Additionally, proteins linked to the metabolism of energy mediated by processes such as photosynthesis or photorespiration are commonly reported as well.
*Carbohydrates*

Within this metabolic group, the process of glycolysis converts glucose into pyruvate, and in the gametophytes under study, we found enzymes, such as ATP-DEPENDENT 6-PHOSPHOFRUCTOKINASE 3 (PFK3), involved in the first reaction, two enzymes participating in glycolysis and gluconeogenesis, FRUCTOSE-BISPHOSPHATE ALDOLASE 3 (FBA3), and others catalyzing the decarboxylation of pyruvate to acetyl-CoA, such as PYRUVATE DEHYDROGENASE E1 COMPONENT SUBUNIT BETA-1 (PDH2). The protein FBA3 reported here and also the protein FBA8 were identified in studies of the fern *C. delgadii* [19]. Linked to pyruvate metabolism, we identified two phosphoenolpyruvate carboxylases (PPC2 and PPC3), which supply oxaloacetate for the tricarboxylic acid cycle, and the protein NAD-DEPENDENT MALIC ENZYME 1 (NAD-ME1), which is involved in regulating the metabolism of sugars and amino acids during the night [35]. Worth mentioning is also 2,3-BIPHOSPHOGLYCERATE-INDEPENDENT PHOSPHOGLYCERATE MUTASE 1 (PGM1), which is important for the functioning of stomatal guard cells and fertility in *A. thaliana* [36]. Gametophytes of *D. affinis* and *D. oreades* produce proteins involved in starch synthesis, including 1,4-ALPHA-GLUCAN-BRANCHING ENZYME 2-2 (SBE2.2) and GRANULE-BOUND STARCH SYNTHASE 1 (GBSS1).
*Tricarboxylic acid cycle and pentose phosphate pathway*

Likewise, we identified some proteins associated with the citrate/tricarboxylic acid cycle, SUCCINATE-CoA LIGASE [ADP-FORMING] SUBUNIT BETA (AT2G20420) and SUCCINATE-CoA LIGASE [ADP-FORMING] SUBUNIT ALPHA-1 (AT5G08300), which are involved in the only phosphorylation step at the substrate level of this cycle. Another protein is MALATE DEHYDROGENASE 1 (MDH1), which catalyzes a reversible NAD-dependent dehydrogenase reaction involved in central metabolism and redox homeostasis between organelle compartments [37]. This protein was also found in three ferns: *P. vittata*, when studying the response to arsenic stress in the roots with or without arbuscular mycorrhizal symbiosis [16]; in germinating spores of *O. cinnamomea* [4]; and in sexual and apomictic gametophytes of *C. thalictroides* [22]. In parallel to glycolysis, the pentose phosphate pathway generates NADPH and pentoses. This metabolic pathway is represented in our dataset by the proteins 6-PHOSPHOGLUCONATE DEHYDROGENASE, 1 DECARBOXYLATING 1 (PGD1), GLUCOSE-6-PHOSPHATE 1-DEHYDROGENASE 6 (G6PD6), and PHOSPHOGLYCERATE KINASE 1 (PGK1). Specifically, a mutation in the gene of the first protein may decrease cellulose synthesis, thus altering the structure and composition of the primary cell wall [38]. The enzyme G6PD6 is important for the synthesis of fatty acids and nucleic acids involved in membrane synthesis and cell division [39]. G6PD6 was also reported in the vegetative tissues of the lycophyte *S. tamariscina*, which was involved in the response to drought [11], and the protein PGD1 was identified by analyzing stipe explants of the fern *C. delgadii*, and found to be associated with direct somatic embryogenesis [19].
*Metabolism of lipids*

Regarding the metabolism of lipids, three proteins were identified in this study. The first protein is ENOYL-[ACYL-CARRIER-PROTEIN] REDUCTASE (MOD1), which catalyzes the last reduction step of the de novo fatty acid synthesis cycle and the fatty acid elongation cycle. A mutation causing a decreased activity of this protein reduces the number of fatty acids, which triggers mosaic premature cell death and changes in the plant’s morphology, such as chlorotic and curly leaves, distorted siliques, and dwarfism [40]. The second protein is ATP-CITRATE SYNTHASE ALPHA CHAIN PROTEIN 1 (ACLA-1), which is necessary for the normal growth and development of plants because it synthesizes acetyl-CoA, a key compound for many metabolic pathways (fatty acids and glucosinolates in chloroplasts; flavonoids, sterols, and phospholipids in the cytoplasm; and ATP and amino acid carbon skeletons in mitochondria). Moreover, it is the substrate for histone acetylation in the nucleus, affecting chromosome structure and regulating transcription [41,42]. The third protein is CITRATE SYNTHASE 2 (CSY2), which synthesizes citrate in peroxisomes for the respiration of fatty acids in seedlings and is required for seed germination [43].
*Biosynthesis of amino acids and nucleotides*

Involved in the biosynthesis of amino acids, we found the proteins ASPARTATE AMINOTRANSFERASE (ASP1); 3-ISOPROPYLMALATE DEHYDRATASE LARGE SUBUNIT (IIL1), which acts in glucosinolate biosynthesis involved in the defense against insects [44]; FERREDOXIN-DEPENDENT GLUTAMATE SYNTHASE 1 (GLU1), which is required for the re-assimilation of ammonium ions generated during photorespiration [45]; HISTIDINOL DEHYDROGENASE (HISN8); and S-ADENOSYLMETHIONINE SYNTHASE 4 (METK4) [46,47,48]. We also identified proteins associated with the metabolism of nucleotides, specifically with AMP syntheses, such as ADENOSINE KINASE 1 (ADK1) and ADENYLOSUCCINATE SYNTHETASE (PURA). Of note is the protein PROBABLE RIBOSE-5-PHOSPHATE ISOMERASE 3 (RPI3), which is essential for the synthesis of numerous compounds such as purines, pyrimidines, aromatic amino acids, NAD, and NADP [38]. Apart from the proteins mentioned above, we found some that are associated with the biosynthesis of nucleotide sugars, such as the two pyrophosphorylases GLUCOSE-1-PHOSPHATE ADENYLYLTRANSFERASE SMALL SUBUNIT (ADG1) and GLUCOSE-1-PHOSPHATE ADENYLYLTRANSFERASE LARGE SUBUNIT 1 (ADG2).
*Metabolism of energy*

The protein OXYGEN-EVOLVING ENHANCER PROTEIN 1-2 (PSBO2), which regulates the replacement of the protein D1 impaired by light [49], and the protein THYLAKOID LUMENAL 16.5 kDa were reported. The latter protein is necessary to carry out photosynthesis correctly and efficiently under two conditions: controlled photoinhibitory light and fluctuating light. In nature, plants experience rapid and extreme changes in sunlight, requiring rapid adaptation [50]. Involved in photosynthesis, we found the protein PHOTOSYNTHETIC NDH SUBUNIT OF LUMENAL LOCATION 5 (PNSL5), which modulates the conformation of the protein BRASSINAZOLE-RESISTANT 1 (BZR1) [51]. This protein binds to the promoter of the gene *FLOWERING LOCUS D* (*FLD*) and represses its expression, eventually leading to the expression of the *FLOWERING LOCUS C* (*FLC*) gene, which encodes a repressor of flowering [51]. Finally, CBBY-LIKE PROTEIN (CBBY) degrades xylulose-1,5-bisphosphate, a potent inhibitor of the protein RUBISCO [52]. On the other hand, photorespiration represents a waste of the energy produced by photosynthesis. The enzyme D-GLYCERATE 3-KINASE (GLYK) catalyzes the final reaction of photorespiration [53]. Another important protein for photorespiration is SERINE-GLYOXYLATE AMINOTRANSFERASE (AGT1), which also participates in primary and lateral root development [54]. The latter protein was also reported in the fern *C. delgadii* [19].
*Sulfur and nitrogen metabolism*

Proteins involved in sulfur metabolism are represented by UDP-SULFOQUINOVOSE SYNTHASE (SQD1), which converts UDP-glucose and sulfite to the precursor of the main group of sulfolipids, UDP-sulfoquinovose, thus preventing sulfite from accumulating as it is toxic to the cell [55], and SUFE-LIKE PROTEIN 1 (SUFE1), a sulfur acceptor that activates cysteine desulfurases in plastids and mitochondria, which is essential for embryogenesis [56]. Regarding nitrogen metabolism, there are the proteins NITROGEN REGULATORY PROTEIN P-II HOMOLOG (GLB1), which is a nitrogen regulatory protein and intervenes in glycosaminoglycan degradation [57], and FERREDOXIN-NITRITE REDUCTASE (NIR1), which catalyzes the reduction of nitrite to ammonium [58]. As the amount of this protein in the cell increases, the tolerance and assimilation of nitrogen dioxide by the plant improves. As nitrogen dioxide is an air pollutant produced largely by motorized vehicles, plants could act as a sink for this substance, i.e., this protein could be used in biotechnological applications for bioremediation [58].
*Metabolism of secondary compounds*

Several proteins related to flavonoid biosynthesis are represented in this work, such as CHALCONE-FLAVANONE ISOMERASE 1 (CHI1), which is responsible for the isomerization of chalcones into naringenin [59]. We also found enzymes involved in the biosynthesis of terpenoids, such as HETERODIMERIC GERANYLGERANYL PYROPHOSPHATE SYNTHASE LARGE SUBUNIT 1 (GGPPS1); the biosynthesis of lignans, PHENYLCOUMARAN BENZYLIC ETHER REDUCTASE 1 (PCBER1); and the biosynthesis of phenylpropanoids. We also found the protein 4-COUMARATE-COA LIGASE 3 (4CL3), which produces CoA-thioesters of hydroxy- and methoxy-substituted cinnamic acids, used to synthesize anthocyanins, flavonoids, isoflavonoids, coumarins, lignin, suberin, and phenols [60], and 3-PHOSPHOSHIKIMATE 1-CARBOXYVINYLTRANSFERASE (AT2G45300), involved in the synthesis of chorismate, which is the precursor of the amino acids phenylalanine, tryptophan, and tyrosine [61]. The proteins 4CL3 and the transferases GLUTATHIONE S-TRANSFERASE L2 (GSTL2) and GLUTATHIONE S-TRANSFERASE L3 (GSTL3) found in our species, was also studied in the fern *A. spinulosa* [29]. The last ones catalyze the glutathione-dependent reduction of S-glutathionyl quercetin to quercetin [62]. Besides, GSTL2 and GSTL3 proteins were also reported in a lycophyte, *S. tamariscina*, where they are required in the response to desiccation [11].

### 3.2. Genetic Information Processing


*Transcription and translation*


In the gametophytes of *D. affinis* and *D. oreades*, we identified two proteins involved in transcription, specifically the 14-3-3-like proteins 14-3-3-LIKE PROTEIN GF14 NU (GRF7) and 14-3-3-LIKE PROTEIN GF14 IOTA (GRF12), which are associated with a DNA-binding complex that binds to the G-box, a cis-regulatory DNA element [63]. These 14-3-3-like proteins were also studied in two species: the lycophyte *S. tamariscina* [11] and the fern *C. delgadii* [19]. Related to translation, we found GLYCINE-RICH RNA-BINDING PROTEIN 3 (RBG3), which has a role in RNA processing during stress, specifically in editing cytosine to uracil in mitochondrial RNA, thereby controlling 6% of all mitochondrial editing sites [64]. This protein was also identified in *S. tamariscina* when studying the response to drought [11], as well as other proteins, such as PUTATIVE PENTATRICOPEPTIDE REPEAT-CONTAINING PROTEIN AT1G03510 (PCMP-E3), POLYADENYLATE-BINDING PROTEIN RBP47B (RBP47B), and UBP1-ASSOCIATED PROTEIN 2A (UBA2A), which regulates mRNAs and stabilizes RNAs in the nucleus [65]. Apart from several ribosomal subunits, there are others linked to translation elongation, like the protein ELONGATION FACTOR 2 (LOS1), which is also involved in the response to cold [66].
*Protein folding and sorting*

Once the proteins have been formed, there is a quality check to ensure that they have been synthesized completely and folded correctly. Among the proteins playing a major role in the acceleration of folding or the degradation of misfolded proteins are CHAPERONIN 1 (CPN10-1) and PEPTIDYL-PROLYL CIS-TRANS ISOMERASE FKBP16-4 (FKBP16-4). The gametophyte of the ferns under study harbor proteins linked to the sorting or transport of molecules within the cell and between the inside and outside of cells. In line with this, we found PROTEIN TRANSLOCASE SUBUNIT SECA1 (SECA1), which has a role in coupling ATP hydrolysis to protein transfer across the thylakoid membrane, thus participating in photosynthetic acclimation and chloroplast formation [67]; IMPORTIN SUBUNIT ALPHA-2 (IMPA2), which acts in nuclear import [68]; the proteins ADP and ATP CARRIER (AAC2), which mediates the import of ADP into the mitochondrial matrix [69], and TIC62 (TIC62), which is involved in the import of nuclear-encoded proteins into chloroplasts [70]. The IMPA-2 protein was also identified when studying germinating spores of the fern *O. cinnamomea* [4]. In addition, we found proteins associated with the transport of water and small hydrophilic molecules through the cell membrane: PROBABLE AQUAPORIN PIP1-4 (PIP1.4) [71]. COATOMER SUBUNIT ALPHA-1 (AT1G62020) and COATOMER SUBUNIT GAMMA (AT4G34450) are associated with clathrin-uncoated vesicles that are transported from the endoplasmic reticulum to the Golgi apparatus and vice versa. In contrast, the proteins CLATHRIN INTERACTOR EPSIN 2 (EPSIN2) and DYNAMIN-2B (DRP2B) are related to clathrin-coated vesicles, with the latter participating in planar polarity formation to correctly positioning the root hairs [72].
*Protein degradation*

Among the proteases, we found ATP-DEPENDENT CLP PROTEASE PROTEOLYTIC SUBUNIT-RELATED PROTEIN 3 (CLPR3) and ATP-DEPENDENT CLP PROTEASE ATP-BINDING SUBUNIT CLPT2 (CLPR2). Plants need to cope with heat stress, and for this, the gametophytes studied here rely on the aminopeptidases LEUCINE AMINOPEPTIDASE 1 and LEUCINE AMINOPEPTIDASE 3 (LAP1 and LAP3), which are probably involved in the processing and turnover of intracellular proteins and function as molecular chaperones protecting proteins from heat-induced damage [73].

### 3.3. Protein–Protein Interactions

Using the STRING platform, we thoroughly analyzed—one by one—the interactions of the groups of proteins studied. We observed that for the metabolism of carbohydrates, evidence from co-expression, text mining, and experiments were stronger between the mitochondrial proteins SUCCINATE-CoA LIGASE [ADP-FORMING] SUBUNIT BETA and SUBUNIT ALPHA-1 than the others in this group. Both proteins are involved in the tricarboxylic acid cycle [74].

Among the proteins for biosynthesis of amino acids, evidence from co-expression was stronger between the proteins ASPARTATE-SEMIALDEHYDE DEHYDROGENASE and DIHYDROXY-ACID DEHYDRATASE (DHAD), while evidence from databases was stronger between DIHYDROXY-ACID DEHYDRATASE and 2-ISOPROPYLMALATE SYNTHASE 2 (IPMS2), 3-ISOPROPYLMALATE DEHYDRATASE LARGE SUBUNIT (IIL1) and 2-ISOPROPYLMALATE SYNTHASE 2, and 3-ISOPROPYLMALATE DEHYDRATASE LARGE SUBUNIT and 3-ISOPROPYLMALATE DEHYDROGENASE 2 (IMD2). In fact, these proteins are involved in the synthesis of numerous compounds necessary for plant growth and development: ASPARTATE-SEMIALDEHYDE DEHYDROGENASE for the biosynthesis of lysine, threonine, and methionine [75]; DIHYDROXY-ACID DEHYDRATASE for isoleucine and valine [76]; 2-ISOPROPYLMALATE SYNTHASE 2 and 3-ISOPROPYLMALATE DEHYDROGENASE 2 for leucine [77,78]; and 3-ISOPROPYLMALATE DEHYDRATASE LARGE SUBUNIT for glucosinolates [44].

For the metabolism of energy, evidence from co-expression was stronger between the proteins ATP SYNTHASE GAMMA CHAIN 1 (ATPC1) and GLYCERALDEHYDE-3-PHOSPHATE DEHYDROGENASE (GAPA-2), while experimental data provided strong evidence for the interaction between PHOTOSYSTEM I P700 CHLOROPHYLL A APOPROTEIN A1 (PSAA) and PHOTOSYSTEM I IRON-SULFUR CENTER (PSAC). In photosynthesis, the C-terminus of PSAC interacts with PSAA and other proteins, such as PHOTOSYSTEM I P700 CHLOROPHYLL A APOPROTEIN A2 (PSAB) and PHOTOSYSTEM I REACTION CENTER SUBUNIT II-1 (PSAD1), for its assembly into the photosystem I [79]. Evidence from databases indicated an interaction between SERINE-GLYOXYLATE AMINOTRANSFERASE (*AGT1*) and GLYCOLATE OXIDASE 2 (GLO2), both proteins being involved in photorespiration [54], while evidence from text mining suggested an interaction between OXYGEN-EVOLVING ENHANCER PROTEIN 1-2 (PSBO2) and OXYGEN-EVOLVING ENHANCER PROTEIN 2-1 (PSBP1), both being chloroplastic oxygen-evolving enhancer proteins that form part of photosystem II [49].

For the metabolism of secondary compounds, evidence from text mining was the strongest, indicating an interaction between PHENYLALANINE AMMONIA-LYASE 1 (PAL1) and PHENYLALANINE AMMONIA-LYASE 4 (PAL4). Both proteins participate in the synthesis from phenylalanine of numerous compounds based on the phenylpropane skeleton, which is fundamental to plant metabolism [80]. 

With respect to transcription and translation, co-expression evidence was stronger between ribosomal proteins. 

Finally, for protein transport, co-expression data provided the strongest evidence for interactions between the proteins COATOMER SUBUNIT ALPHA-1 and COATOMER SUBUNIT GAMMA; experimental data for the interaction between COATOMER SUBUNIT ALPHA-1 and COATOMER SUBUNIT DELTA; and text mining data for the interactions between PROTEIN TRANSLOCASE SUBUNIT SECA1 and ATPase GET3B.

As indicated in the results, in the group related to the metabolism of carbohydrates, the protein with the most interactions was PHOSPHOGLYCERATE KINASE 1, which is involved in glycolysis [81]. For the biosynthesis of amino acids, ASPARTATE-SEMIALDEHYDE DEHYDROGENASE, DIHYDROXY-ACID DEHYDRATASE, and 3-ISOPROPYLMALATE DEHYDROGENASE 2 were involved in several biosynthetic pathways: lysine, leucine, valine, isoleucine, methionine, and threonine [77]. In the group metabolism of energy, ATP SYNTHASE GAMMA CHAIN 1 had the highest number of interactions, likely because it is part of a chloroplastic ATP synthase [82]. The protein 4-COUMARATE-COA LIGASE 3 had the most interactions in the group metabolism of secondary compounds. It plays a key role in the synthesis of numerous secondary metabolites, such as anthocyanins, flavonoids, isoflavonoids, coumarins, lignin, suberin, and phenols [60]. In transcription and translation, the ribosomal proteins LARGE RIBOSOMAL SUBUNIT PROTEIN UL4Z, SMALL RIBOSOMAL SUBUNIT PROTEIN US11X, and SMALL RIBOSOMAL SUBUNIT PROTEIN US17Y, which are necessary for the formation of ribosomes, had the highest number of interactions [83]. Finally, in transport, the proteins PROTEIN TRANSLOCASE SUBUNIT SECA1 and COATOMER SUBUNIT GAMMA participate in coupling ATP hydrolysis to protein transfer across the thylakoid membrane and in the transport of clathrin-uncoated vesicles from the endoplasmic reticulum to the Golgi apparatus and vice versa, respectively [67].

Regarding the statistical analysis of the two highest scoring types of interactions in the studied groups of metabolism of carbohydrates (database and text mining) and transcription and translation (experiments and co-expression), Pearson’s correlation coefficients, which measure the tendency of two vectors to increase or decrease together, were significant. One of the most popular types of data in databases is text, and the process of synthesizing information is known as text mining. In the case of proteins linked to carbohydrate metabolism, there seems to be a lot of information in databases about it, and therefore text mining could be enriched as well. Regarding transcription and translation, we speculate that most of the experiments on molecular biology cope with these processes, reporting more genes involved on them.

## 4. Materials and Methods

### 4.1. Plant Material and Growth Conditions

Spores of *D. affinis* were obtained from sporophytes growing in Turón valley (Asturias, Spain), 477 m a.s.l., 43°12′10″ N−5°43′43″ W. For *D. oreades*, spores were collected from sporophytes growing in Neila lagoons (Burgos, Spain), 1.920 m a.s.l., 42°02′48″ N−3°03′44″ W. Spores were released from sporangia, soaked in water for 2 h, and then washed for 10 min with a solution of NaClO (0.5%) and Tween 20 (0.1%). Then, they were rinsed three times with sterile, distilled water. Spores were centrifuged at 1300× *g* for 3 min between rinses and then cultured in 500 mL Erlenmeyer flasks containing 100 mL of liquid Murashige and Skoog (MS) medium [84]. Unless otherwise noted, media were supplemented with 2% sucrose (*w*/*v*), and the pH was adjusted to 5.7 with 1 or 0.1 N NaOH. The cultures were kept on an orbital shaker (75 rpm) at 25 °C under cool, white fluorescent light (70 µmol m^−2^s^−1^) with a 16 h photoperiod.

Following spore germination, filamentous gametophytes were subcultured into 200 mL flasks containing 25 mL of MS medium supplemented with 2% sucrose (*w*/*v*) and 0.7% agar. The gametophytes of *D. affinis* became two-dimensional, arriving at the spatulate and heart stage after 20 or 30 additional days, respectively. Gametophytes of *D. oreades* grew slower and needed around six months to become cordate and reach sexual maturity (Figure 5). Apomictic and sexual gametophytes were collected, and images were taken under a light microscope (Nikon Eclipse E600, Tokyo, Japan) using microphotographic equipment (DS Camera Control, Nikon, Tokyo, Japan). Gametophytes of *D. oreades* had only female reproductive organs (i.e., archegonia), while cordate, apomictic gametophytes of *D. affinis* had visible developing apogamic centers composed of smaller and darker isodiametric cells. Samples of apomictic and sexual cordate gametophytes were weighed before and after lyophilization for 48 h (Telstar-Cryodos, Terrassa, Spain) and stored in Eppendorf tubes in a freezer at −20 °C until use.

### 4.2. Protein Extraction, Separation, and In-Gel Digestion

The present work expands previous bioinformatic analyses, and, specifically, it is carried out with a set of 218 proteins, which had been extracted and annotated as follows.

The protocol used for protein extraction, separation, and in-gel digestion was reported earlier [31]. In brief, samples were solubilized with 800 μL of buffer A (0.5 M Tris-HCl (pH 8.0), 5 mM EDTA, 0.1 MHEPES-KOH, 4 mM DTT, 15 mM EGTA, 1 mM PMSF, 0.5% (*w*/*v*) PVP, and 1x protease inhibitor cocktail (Roche, Rotkreuz, Switzerland)) and homogenized, and proteins were extracted in two steps: first, the homogenate was subjected to centrifugation at 16,200× *g* for 10 min at 4 °C on a tabletop centrifuge, and, second, the supernatant was subjected to ultracentrifugation at 117–124 kPa (100,000× *g*) for 45 min at 4 °C in an Airfuge (Beckman Coulter, Pasadena, CA, USA), yielding the soluble protein fraction in the supernatant. In parallel, the pellet obtained from the first ultracentrifugation was re-dissolved in 200 μL of buffer B (40 mM Tris-base, 40 mM DTT, 4% (*w*/*v*) SDS, and 1× protease inhibitor cocktail (Roche, Rotkreuz, Switzerland)) to extract membrane proteins using the ultracentrifuge, as described above, in the supernatant. Protein concentrations were determined using a Qubit Fluorometer (Invitrogen, Carlsbad, CA, USA), and 1D gel electrophoresis was performed as follows: 1 mg protein was treated with sample loading buffer and 2 M DTT, heated at 99 °C for 5 min, followed by a short cooling period on ice, and then loaded separately onto a 0.75 mm thick, 12% SDS-PAGE mini-gel. Electrophoresis conditions were 150 V and 250 mA for 1 h in 1× running buffer.

### 4.3. Protein Separation and In-Gel Digestion

Each gel lane was cut into six 0.4 cm wide sections resulting in 48 slices, and then fragmented into smaller pieces and subjected to 10 mM DTT (in 25 mM AmBic, pH8) for 45 min at 56 °C and 50 mM iodoacetamide for 1 h at room temperature in the dark prior to trypsin digestion at 37 °C overnight. Subsequently, gel pieces were washed twice with 100 μL of 100 mM NH4HCO_3_/50% acetonitrile and washed once with 50 μL acetonitrile. At this point, the supernatants were discarded. Peptides were digested with 20 μL trypsin (5 ng/L in 10 mM Tris/2 mM CaCl_2_, pH 8.2) and 50 μL buffer (10 mM Tris/2 mM CaCl_2_, pH 8.2). After microwave-heating for 30 min at 60 °C, the supernatant was removed, and gel pieces were extracted once with 150 μL 0.1% TFA/50% acetonitrile. All supernatants were put together, then dried and dissolved in 15 μL 0.1% formic acid/3% acetonitrile, and, finally, transferred to auto-sampler vials for liquid chromatography (LC)-tandem mass spectrometry (MS/MS) for which 5 μL was injected.

### 4.4. Protein Identification, Verification, and Bioinformatic Downstream Analyses

MS/MS and peptide identification (Orbitrap XL) were performed according to [31]. Scaffold software (version Scaffold 4.2.1, Proteome Software Inc., Portland, OR, USA) was used to validate MS/MS-based peptide and protein identifications. Mascot results were analyzed together using the MudPIT option. Peptide identifications were accepted if they scored better than 95.0% probability as specified by the Peptide Prophet algorithm with delta mass correction, and protein identifications were accepted if the Protein Prophet probability was above 95%. Indeed, the unique peptide ≥ 2′ was considered. Proteins that contained the same peptides and could not be differentiated based on MS/MS alone were grouped to satisfy the principles of parsimony using the scaffolds cluster analysis option. Only proteins that met the above criteria were considered as positively identified for further analysis. The number of random matches was evaluated by performing the Mascot searches against a database containing decoy entries and checking how many decoy entries (proteins or peptides) passed the applied quality filters. The peptide FDR and protein FDR were estimated at 2% and 1%, respectively, indicating the stringency of the analyses. A semi-quantitative spectrum counting analysis was conducted. The “total spectrum count” for each protein and each sample was reported, and these spectrum counts were averaged for each species, *D. affinis* and *D. oreades*. Then, one “1” was added to each average in order to prevent division by zero, and a log2 ratio of the averaged spectral counts from *D. affinis* versus *D. oreades* was calculated. Proteins were considered as differentially expressed if this log2 ratio was above 0.99. This refers to at least twice as many peptide spectrum match (PSM) assignments in one group compared to the other. Also, to provide some functional understanding of the identified proteins, we blasted the whole protein sequences of all identified proteins against *Sellginella moellendorfii* and *A. thaliana* Uniprot sequences and retrieved the best matching identifier from each of them along with the corresponding e-value, accepting blast-hits with values below 1 × 10^−7^. These better-described ortholog identifiers were then used in further downstream analysis.

### 4.5. Protein Analysis Using the STRING Platform

The identifiers of the genes from apomictic and sexual gametophyte samples were used as input for STRING platform version 11.5 analysis, and a high threshold (0.700) was selected for a positive interaction between a pair of proteins.

### 4.6. Statistical Analyses

Regarding the two major protein–protein interactions highlighted for carbohydrate metabolism, amino acid biosynthesis, energy metabolism, and transcription and translation, a Pearson’s correlation test was performed using R software 4.2.0, and *p*-values lower than 0.05 were considered significant.

## 5. Conclusions

The analysis of a set of 218 proteins shared by the gametophytes of the apomictic fern *D. affinis* and its sexual relative *D. oreades* revealed the presence of proteins mostly involved in biological functions associated with metabolism, the processing of genetic information, and abiotic stresses. Some smaller protein groups were studied in detail: metabolism of carbohydrates; biosynthesis of amino acids; metabolism of energy; metabolism of secondary compounds; transcription, translation, and transport; and abiotic stress. Possible interactions between these proteins were identified, with the most common source of evidence for interactions stemming from databases and information from text mining. The proteins involved in transcription and translation exhibit the strongest interactions. The description of possible biological functions and the possible protein–protein interactions among the identified proteins expands our current knowledge about ferns and plants in general.

## Figures and Tables

**Figure 1 ijms-24-12429-f001:**
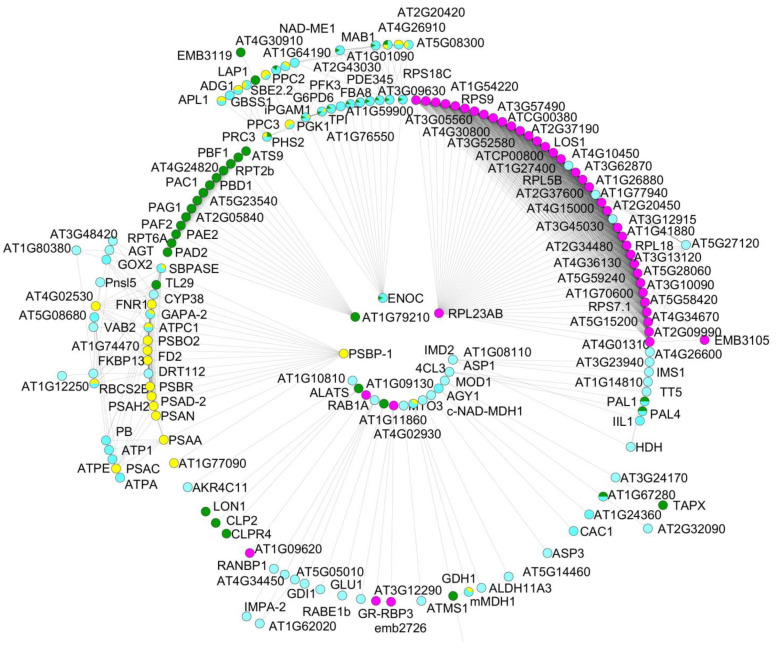
GO enrichment terms of the proteomes shared by gametophytes of *D. affinis* and *D. oreades* according to the category biological function; analyzed by STRING and CYTOSCAPE. Turquoise indicates metabolism of carbohydrates, yellow metabolism of energy, pink transcription and translation, and green protein degradation.

**Figure 2 ijms-24-12429-f002:**
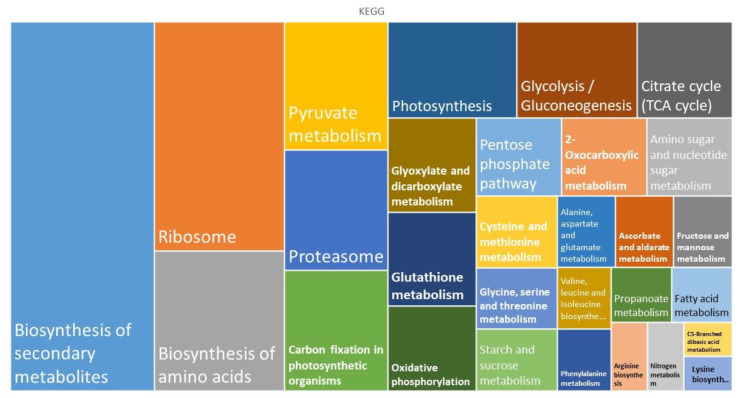
KEGG enrichment terms of the proteomes shared by gametophytes of *D. affinis* and *D. oreades;* analyzed using the STRING platform.

**Figure 3 ijms-24-12429-f003:**
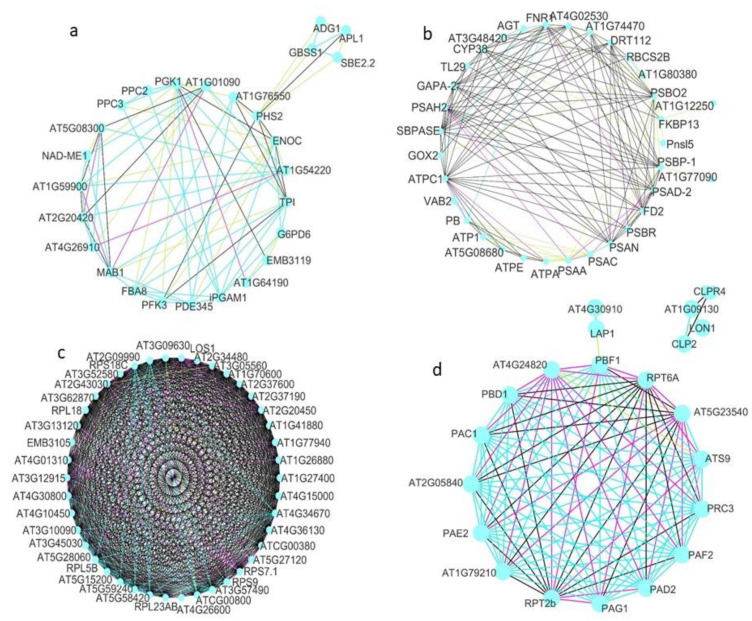
Circular representations obtained by STRING and CYTOSCAPE for proteins detected in the gametophytes of both *D. affinis* and *D. oreades*: (**a**) metabolism of carbohydrates, (**b**) metabolism of energy, (**c**) ribogenesis, and (**d**) protein degradation. Pink lines refer to evidence from experiments, green lines from text mining, black lines from co-expression, and blue lines from databases.

**Figure 4 ijms-24-12429-f004:**
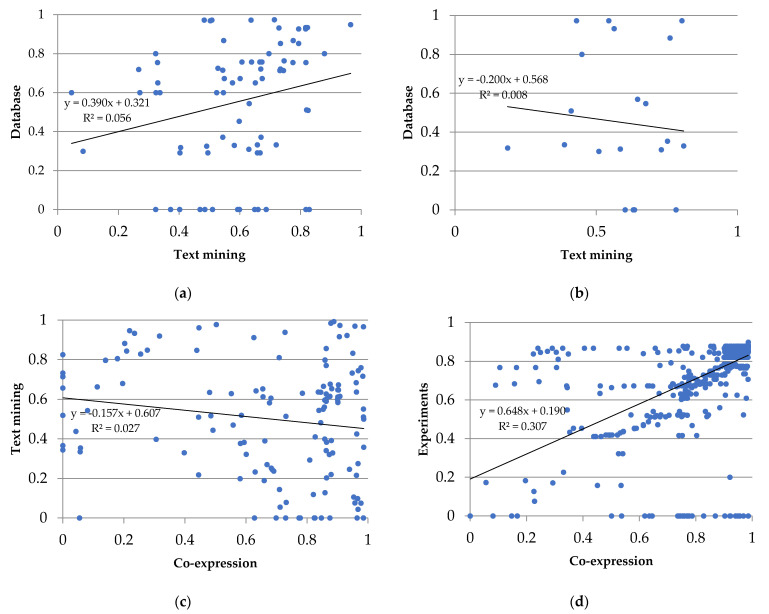
Plots of the two main types of evidence for interactions in the groups of proteins shared by the gametophytes of *D. affinis* and *D. oreades*: (**a**) metabolism of carbohydrates, (**b**) biosynthesis of amino acids, (**c**) metabolism of energy, and (**d**) transcription and translation. Each spot represents the intersection of the type of evidence for interactions between two proteins. The linear regression and the coefficient of correlation are provided for each pair of evidence for the interaction.

**Figure 5 ijms-24-12429-f005:**
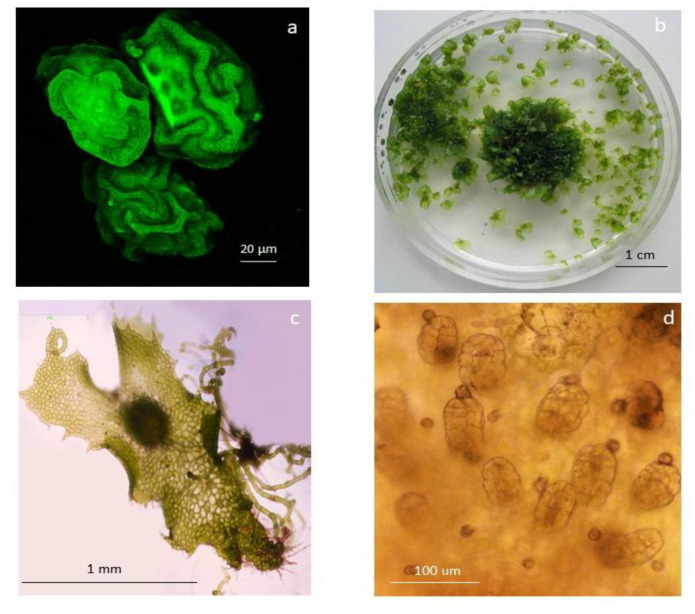
Morphological traits in the apomictic fern *D. affinis* and its sexual relative *D. oreades*: (**a**) confocal image of spores of *D. affinis*; (**b**) gametophytes of *D. affinis* growing in a Petri dish; (**c**) image taken under a light microscope of a chordate gametophyte of *D. affinis* showing an apomictic center in the middle; and (**d**) archegonia in the gametophyte of *D. oreades*.

**Table 1 ijms-24-12429-t001:** Proteins involved in ribogenesis and protein degradation found in the gametophyte of the ferns *D. affinis* and *D. oreades*.

Category	Accession Number	UniProtKB/ Swiss-Prot	Gene Name	Protein Name	MW (kDa)	% Coverage	Exclusive Unique Peptides	Total Spectrum	E-Value
Ribogenesis	1-12177_3_ORF5	P56799	*RPS4*	SMALL RIBOSOMAL SUBUNIT PROTEIN US4C	25.6	0	0	3	7.63 × 10^−70^
Ribogenesis	16646-605_4_ORF2	Q9SX68	*RPL18*	LARGE RIBOSOMAL SUBUNIT PROTEIN UL18C	20.9	7	1	3	7.35 × 10^−57^
Ribogenesis	24557-506_3_ORF2	Q9SKX4	*RPL3A*	LARGE RIBOSOMAL SUBUNIT PROTEIN UL3C	29.3	14	2	9	5.37 × 10^−121^
Ribogenesis	139931-207_5_ORF1	O04603	*RPL5*	LARGE RIBOSOMAL SUBUNIT PROTEIN UL5C	32.4	8	2	14	2.41 × 10^−92^
Ribogenesis	26987-486_4_ORF2	Q9M3C3	*RPL23AB*	LARGE RIBOSOMAL SUBUNIT PROTEIN UL23Y	20.2	31	1	39	3.98 × 10^−60^
Ribogenesis	6809-878_4_ORF2	P49227	*RPL5B*	LARGE RIBOSOMAL SUBUNIT PROTEIN UL18Y	34.7	13	4	12	1.09 × 10^−53^
Ribogenesis	87113-269_4_ORF1	Q9M9W1	*RPL22B*	LARGE RIBOSOMAL SUBUNIT PROTEIN EL22Z	16.1	14	1	3	1.35 × 10^−52^
Ribogenesis	21899-533_1_ORF2	P51419	*RPL27C*	LARGE RIBOSOMAL SUBUNIT PROTEIN EL27X	18.2	19	3	22	7.19 × 10^−65^
Ribogenesis	24282-509_2_ORF1	P50883	*RPL12A*	LARGE RIBOSOMAL SUBUNIT PROTEIN UL11Z	19.1	13	0	6	1.24 × 10^−104^
Ribogenesis	36225-421_1_ORF2	Q9SIM4	*RPL14A*	LARGE RIBOSOMAL SUBUNIT PROTEIN EL14Z	17.9	19	2	11	2.62 × 10^−68^
Ribogenesis	10983-723_3_ORF2	P51418	*RPL18AB*	LARGE RIBOSOMAL SUBUNIT PROTEIN EL20Y	22.9	10	1	9	1.99 × 10^−119^
Ribogenesis	34727-430_6_ORF2	P49637	*RPL27AC*	LARGE RIBOSOMAL SUBUNIT PROTEIN UL15X	16.3	7	1	4	8.73 × 10^−78^
Ribogenesis	73912-293_6_ORF2	Q42064	*RPL8C*	LARGE RIBOSOMAL SUBUNIT PROTEIN UL2X	28.7	25	4	42	2.24 × 10^−167^
Ribogenesis	37807-413_3_ORF1	Q93VI3	*RPL17A*	LARGE RIBOSOMAL SUBUNIT PROTEIN UL22Z	23.5	4	1	7	5.22 × 10^−107^
Ribogenesis	176389-160_3_ORF2	Q42351	*RPL34A*	LARGE RIBOSOMAL SUBUNIT PROTEIN EL34Z	18.2	12	1	15	1.23 × 10^−65^
Ribogenesis	8788-797_5_ORF2	Q9FZH0	*RPL35AB*	LARGE RIBOSOMAL SUBUNIT PROTEIN EL33Z	13.6	18	0	17	6.54 × 10^−60^
Ribogenesis	27503-481_2_ORF2	O80929	*RPL36A*	LARGE RIBOSOMAL SUBUNIT PROTEIN EL36Z	13	16	1	28	8.82 × 10^−52^
Ribogenesis	75664-290_4_ORF2	Q9SF40	*RPL4A*	LARGE RIBOSOMAL SUBUNIT PROTEIN UL4Z	46.3	12	3	20	0
Ribogenesis	45247-378_5_ORF2	Q9SZX9	*RPL9D*	LARGE RIBOSOMAL SUBUNIT PROTEIN UL6X	25.1	7	1	11	1.63 × 10^−105^
Ribogenesis	86488-270_3_ORF2	Q9LZH9	*RPL7AB*	LARGE RIBOSOMAL SUBUNIT PROTEIN EL8Y	32.5	26	8	33	7.63 × 10^−154^
Ribogenesis	163051-176_2_ORF2	Q8VZ19	*RPL30B*	LARGE RIBOSOMAL SUBUNIT PROTEIN EL30Y	16.7	13	2	9	1.91 × 10^−61^
Ribogenesis	69050-304_1_ORF1	P49200	*RPS20A*	SMALL RIBOSOMAL SUBUNIT PROTEIN US10Z	21.1	12	2	14	3.32 × 10^−72^
Ribogenesis	5816-941_6_ORF2	P42036	*RPS14C*	SMALL RIBOSOMAL SUBUNIT PROTEIN US11X	18.7	22	3	7	2.52 × 10^−87^
Ribogenesis	39126-407_1_ORF2	Q8LC83	*RPS24B*	SMALL RIBOSOMAL SUBUNIT PROTEIN ES24Y	20.7	7	1	2	8.38 × 10^−74^
Ribogenesis	108940-238_6_ORF2	Q42262	*RPS3AB*	SMALL RIBOSOMAL SUBUNIT PROTEIN ES1Y	32.7	22	5	21	8.87 × 10^−155^
Ribogenesis	85164-272_6_ORF2	Q8VYK6	*RPS4D*	SMALL RIBOSOMAL SUBUNIT PROTEIN ES4X	30	7	2	18	9.28 × 10^−165^
Ribogenesis	45770-376_1_ORF2	Q9LXG1	*RPS9B*	SMALL RIBOSOMAL SUBUNIT PROTEIN US4Z	24.7	4	0	9	1.32 × 10^−121^
Ribogenesis	140134-206_2_ORF2	F4JB06	*MGH6.2*	RIBOSOMAL PROTEIN S5/ELONGATION FACTOR G/III/V FAMILY PROTEIN	16.9	7	0	1	2 × 10^−50^
Ribogenesis	1627-1498_1_ORF1	P61841	*RPS7-A*	SMALL RIBOSOMAL SUBUNIT PROTEIN US7CZ	18.8	13	3	11	3.24 × 10^−77^
Ribogenesis	31704-450_4_ORF1	Q9FIF3	*ES8Y*	RIBOSOMAL PROTEIN ES8Y	16.8	28	1	19	1.93 × 10^−73^
Ribogenesis	11320-714_2_ORF2	Q9XJ27	*RPS9*	SMALL RIBOSOMAL SUBUNIT PROTEIN US9C	25.9	3	1	3	1.03 × 10^−76^
Ribogenesis	21394-539_3_ORF2	P42791	*RPL18B*	LARGE RIBOSOMAL SUBUNIT PROTEIN EL18Y	23.4	25	3	21	1.03 × 10^−105^
Ribogenesis	40578-399_1_ORF2	P56798	*RPS3*	SMALL RIBOSOMAL SUBUNIT PROTEIN US3C	25.3	6	1	6	6.73 × 10^−82^
Ribogenesis	10791-728_3_ORF2	O65569	*RPS11B*	SMALL RIBOSOMAL SUBUNIT PROTEIN US17Y	20.5	12	2	36	5.17 × 10^−84^
Ribogenesis	66444-310_2_ORF2	Q1PEP5	*NUCL2*	NUCLEOLIN 2	61.5	12	6	11	6.83 × 10^−49^
Ribogenesis	260531-93_3_ORF2	O04658	*NOP5-1*	PROBABLE NUCLEOLAR PROTEIN 5-1	62.5	11	5	10	0
Proteasome	93086-259_4_ORF1	Q9LT08	*RPN11*	26S PROTEASOME NON-ATPASE REGULATORY SUBUNIT 14 HOMOLOG	34.9	3	1	5	0
Proteasome	78122-285_4_ORF2	Q93Y35	*RPN7*	26S PROTEASOME NON-ATPASE REGULATORY SUBUNIT 6 HOMOLOG	44.8	4	1	9	0
Proteasome	353924-42_4_ORF1	O23712	*PAF2*	PROTEASOME SUBUNIT ALPHA TYPE-1-B	13,9	8	1	2	2.8 × 10^−136^
Proteasome	7073-864_3_ORF1	O23708	*PAB1*	PROTEASOME SUBUNIT ALPHA TYPE-2-A	26.4	9	2	10	4.87 × 10^−158^
Proteasome	326729-54_1_ORF2	O81148	*PAC1*	PROTEASOME SUBUNIT ALPHA TYPE-4-A	30.6	13	3	9	3.69 × 10^−154^
Proteasome	68626-305_4_ORF2	Q42134	*PAE2*	PROTEASOME SUBUNIT ALPHA TYPE-5-B	28.7	28	3	13	4.68 × 10^−158^
Proteasome	23928-513_3_ORF1	O81147	*PAA2*	PROTEASOME SUBUNIT ALPHA TYPE-6-B	30.8	7	3	12	9.67 × 10^−153^
Proteasome	340935-47_4_ORF1	O24616	*PAD2*	PROTEASOME SUBUNIT ALPHA TYPE-7-B	27.4	18	4	11	3.86 × 10^−148^
Proteasome	14462-637_1_ORF2	O23714	*PBD1*	PROTEASOME SUBUNIT BETA TYPE-2-A	24.3	4	1	7	1.43 × 10^−103^

**Table 2 ijms-24-12429-t002:** Biological functions and number of interactions exhibited by proteins common to apomictic *D. affinis* and sexual *D. oreades*.

Biological Function	Protein	Nº of Interactions
Metabolism of carbohydrates	PHOSPHOGLYCERATE KINASE 1	11
Biosynthesis of amino acids	3-ISOPROPYLMALATE DEHYDROGENASE	5
	DIHYDROXY-ACID DEHYDRATASE	5
	ASPARTATE-SEMIALDEHYDE DEHYDROGENASE	5
Metabolism of energy	ATP SYNTHASE GAMMA CHAIN 1	18
Secondary metabolism	4-COUMARATE-COA LIGASE 3	3
Transcription & Translation	LARGE RIBOSOMAL SUBUNIT PROTEIN UL4Z	44
	SMALL RIBOSOMAL SUBUNIT PROTEIN US11X	44
	SMALL RIBOSOMAL SUBUNIT PROTEIN US17Y	44
Transport	PROTEIN TRANSLOCASE SUBUNIT SECA1	2
	COATOMER SUBUNIT GAMMA	2

**Table 3 ijms-24-12429-t003:** Selected proteins found in gametophytes of both apomictic *D. affinis* and sexual *D. oreades*.

Category	Accession Number	UniProtKB/ Swiss-Prot	Gene Name	Protein Name	MW (kDa)	% Coverage	Exclusive Unique Peptides	Total Spectra	E-Value
Carbohydrates	58787-330_2_ORF2	Q94AA4	*PFK3*	ATP-DEPENDENT 6-PHOSPHOFRUCTOKINASE 3	64.5	2	1	1	0
Carbohydrates	135690-210_1_ORF2	Q9ZU52	*FBA3*	FRUCTOSE-BISPHOSPHATE ALDOLASE 3	42.7	32	7	38	0
Carbohydrates	38153-411_5_ORF2	Q38799	*PDH2*	PYRUVATE DEHYDROGENASE E1 COMPONENT SUBUNIT BETA-1	40.3	14	4	6	0
Carbohydrates	83096-276_3_ORF2	Q5GM68	*PPC2*	PHOSPHOENOLPYRUVATE CARBOXYLASE 2	112.8	6	5	5	0
Carbohydrates	54280-344_1_ORF1	Q84VW9	*PPC3*	PHOSPHOENOLPYRUVATE CARBOXYLASE 3	111.8	1	0	1	0
Carbohydrates	113756-233_2_ORF1	Q9SIU0	*NAD-ME1*	NAD-DEPENDENT MALIC ENZYME 1	71.4	3	2	3	0
Carbohydrates	102811-246_6_ORF2	O04499	*PGM1*	2,3-BIPHOSPHOGLYCERATE-INDEPENDENT PHOSPHOGLYCERATE MUTASE 1	63.2	4	1	2	0
Carbohydrates	64100-316_1_ORF1	O82662	*AT2G20420*	SUCCINATE-COA LIGASE [ADP-FORMING] SUBUNIT BETA	50	9	0	4	0
Carbohydrates	8279-816_3_ORF2	P68209	*AT5G08300*	SUCCINATE-COA LIGASE [ADP-FORMING] SUBUNIT ALPHA-1	34.6	9	2	3	0
Carbohydrates	222487-119_2_ORF2	P93819	*MDH1*	MALATE DEHYDROGENASE 1	38.4	20	2	16	0
Carbohydrates	156827-185_4_ORF1	Q9SH69	*PGD1*	6-PHOSPHOGLUCONATE DEHYDROGENASE, DECARBOXYLATING 1	58.8	16	2	9	1.37 × 10^−112^
Carbohydrates	12493-682_6_ORF2	Q9FJI5	*G6PD6*	GLUCOSE-6-PHOSPHATE 1-DEHYDROGENASE 6	65.1	6	3	3	0
Carbohydrates	20760-547_4_ORF1	Q9LD57	*PGK1*	PHOSPHOGLYCERATE KINASE 1	19.5	14	0	3	5.63 × 10^−43^
Carbohydrates	69882-302_6_ORF2	Q9LZS3	*SBE2.2*	1,4-ALPHA-GLUCAN-BRANCHING ENZYME 2-2	98.6	5	4	6	0
Carbohydrates	96049-255_6_ORF1	Q9MAQ0	*GBSS1*	GRANULE BOUND STARCH SYNTHASE 1	70.16	1	1	1	0
Lipids	20213-554_2_ORF1	Q9SLA8	*MOD1*	ENOYL-[ACYL-CARRIER-PROTEIN] REDUCTASE [NADH]	41.8	7	1	6	4.48 × 10^−180^
Lipids	387953-27_4_ORF1	Q9SGY2	*ACLA-1*	ATP-CITRATE SYNTHASE ALPHA CHAIN PROTEIN 1	46.8	10	2	4	0
Lipids	211149-128_1_ORF1	Q9LXS6	*CSY2*	CITRATE SYNTHASE 2	57.9	2	0	1	0
Amino acids	47558-369_4_ORF2	P46643	*ASP1*	ASPARTATE AMINOTRANSFERASE	51.5	5	0	5	0
Amino acids	72506-296_4_ORF1	Q94AR8	*IIL1*	3-ISOPROPYLMALATE DEHYDRATASE LARGE SUBUNIT	57.6	6	3	4	0
Amino acids	125905-219_3_ORF2	Q9ZNZ7	*GLU1*	FERREDOXIN-DEPENDENT GLUTAMATE SYNTHASE 1	181.3	6	5	13	0
Amino acids	14065-645_3_ORF2	Q9C5U8	*HISN8*	HISTIDINOL DEHYDROGENASE	55.2	1	2	1	0
Amino acids	294436-71_4_ORF2	Q9LUT2	*METK4*	S-ADENOSYLMETHIONINE SYNTHASE 4	42.7	5	0	2	0
Nucleotides	2121-1366_3_ORF2	Q9SF85	*ADK1*	ADENOSINE KINASE 1	39.2	14	3	4	0
Nucleotides	59309-329_5_ORF1	Q96529	*PURA*	ADENYLOSUCCINATE SYNTHETASE	57.9	2	1	1	0
Nucleotides	152024-193_3_ORF2	Q9S726	*RPI3*	PROBABLE RIBOSE-5-PHOSPHATE ISOMERASE 3	36.1	2	0	2	5.33 × 10^−120^
Nucleotides	27769-479_3_ORF1	P55228	*ADG1*	GLUCOSE-1-PHOSPHATE ADENYLYLTRANSFERASE SMALL SUBUNIT	12.5	7	0	1	4.43 × 10^−63^
Nucleotides	181563-155_3_ORF2	P55229	*ADG2*	GLUCOSE-1-PHOSPHATE ADENYLYLTRANSFERASE LARGE SUBUNIT 1	57.4	6	2	4	0
Energy	154679-189_1_ORF2	Q9S841	*PSBO2*	OXYGEN-EVOLVING ENHANCER PROTEIN 1-2	35.3	35	7	24	6.62 × 10^−141^
Energy	218625-122_1_ORF2	O22773	*AT4G02530*	THYLAKOID LUMENAL 16.5 kDa PROTEIN	24.7	5	1	1	8.55 × 10^−47^
**Category**	**Accession Number**	**UniProtKB/** **Swiss-Prot**	**Gene Name**	**Protein Name**	**MW (kDa)**	**%** **Coverage**	**Exclusive Unique Peptides**	**Total Spectra**	**E-Value**
Energy	6036-926_2_ORF1	Q9ASS6	*PNSL5*	PHOTOSYNTHETIC NDH SUBUNIT OF LUMENAL LOCATION 5	32.2	15	4	10	2.45 × 10^−93^
Energy	250817-99_2_ORF2	Q94K71	*CBBY*	CBBY-LIKE PROTEIN	34.9	7	2	3	6.24 × 10^−131^
Energy	235330-110_2_ORF1	Q944I4	*GLYK*	D-GLYCERATE 3-KINASE	43.9	3	0	1	2.82 × 10^−153^
Energy	297118-70_2_ORF2	Q56YA5	*AGT1*	SERINE-GLYOXYLATE AMINOTRANSFERASE	47.8	2	0	2	0
S&N metabolism	33137-439_6_ORF2	O48917	*SQD1*	UDP-SULFOQUINOVOSE SYNTHASE	54.9	10	3	4	0
S&N metabolism	227095-115_1_ORF2	Q84W65	*SUFE1*	SUFE-LIKE PROTEIN 1	40.7	2	1	1	3.22 × 10^−106^
S&N metabolim	311596-62_2_ORF2	Q9ZST4	*GLB1*	NITROGEN REGULATORY PROTEIN P-II HOMOLOG	23.4	15	1	1	3.28 × 10^−62^
S&N metabolim	318906-58_1_ORF1	Q39161	*NIR1*	FERREDOXIN-NITRITE REDUCTASE	69.6	7	4	8	0
Secondary compounds	156331-186_3_ORF2	P41088	*CHI1*	CHALCONE-FLAVANONE ISOMERASE 1	26.2	6	0	2	6.89 × 10^−56^
Secondary compounds	230420-113_2_ORF2	P34802	*GGPPS1*	HETERODIMERIC GERANYLGERANYL PYROPHOSPHATE SYNTHASE LARGE SUBUNIT 1	41	5	1	1	1.48 × 10^−147^
Secondary compounds	85783-271_1_ORF2	Q9T030	*PCBER1*	PHENYLCOUMARAN BENZYLIC ETHER REDUCTASE 1	34.9	33	9	23	4.03 × 10^−115^
Secondary compounds	156554-185_2_ORF1	Q9S777	*4CL3*	4-COUMARATE-COA LIGASE 3	51.9	2	1	2	0
Secondary compounds	223603-118_1_ORF1	P05466	*AT2G45300*	3-PHOSPHOSHIKIMATE 1-CARBOXYVINYLTRANSFERASE	44.3	2	1	1	0
Oxido -reduction	133847-212_2_ORF2	Q9SID3	*AT2G31350*	HYDROXYACYLGLUTATHIONE HYDROLASE 2	33.1	6	1	1	6.51 × 10^−131^
Oxido -reduction	34437-432_2_ORF1	Q9M2W2	*GSTL2*	GLUTATHIONE S-TRANSFERASE L2	16.1	15	2	5	3.24 × 10^−29^
Oxido -reduction	115571-230_4_ORF1	Q9LZ06	*GSTL3*	GLUTATHIONE S-TRANSFERASE L3	35.3	1	2	1	6.13 × 10^−75^
Transcription	181200-155_2_ORF2	Q96300	*GRF7*	14-3-3-LIKE PROTEIN GF14 NU	32.8	12	1	9	1.11 × 10^−161^
Transcription	287872-75_1_ORF1	Q9C5W6	*GRF12*	14-3-3-LIKE PROTEIN GF14 IOTA	32.8	12	1	9	3.83 × 10^−151^
Translation	209284-130_2_ORF2	Q9FNR1	*RBG3*	GLYCINE-RICH RNA-BINDING PROTEIN 3	19.8	15	2	4	1.45 × 10^−31^
Translation	293356-72_1_ORF1	Q9LR72	*PCMP-E3*	PUTATIVE PENTATRICOPEPTIDE REPEAT-CONTAINING PROTEIN AT1G03510 (POLIPASA)	26.1	5	1	1	0.4
Translation	26795-487_6_ORF2	Q0WW84	*RBP47B*	POLYADENYLATE-BINDING PROTEIN RBP47B	45.9	2	0	1	3.27 × 10^−136^
Translation	174433-162_1_ORF1	Q9ASR1	*LOS1*	ELONGATION FACTOR 2	73.8	9	1	12	0
Folding	26640-489_1_ORF2	Q9M1C2	*CPN10-1*	10 kDa CHAPERONIN 1	19.4	15	2	6	9.48 × 10^−39^
Folding	189606-147_1_ORF2	Q9SR70	*FKBP16-4*	PEPTIDYL-PROLYL CIS-TRANS ISOMERASE FKBP16-4	26.4	13	3	6	1.35 × 10^−89^
Folding	2524-1285_6_ORF2	Q9SKQ0	*CYP19-2*	PEPTIDYL-PROLYL CIS-TRANS ISOMERASE CYP19-2	21.6	27	5	24	2.24 × 10^−90^
Sorting	19573-562_5_ORF2	Q9SYI0	*SECA1*	PROTEIN TRANSLOCASE SUBUNIT SECA1	115.7	2	1	2	0
Sorting	146969-201_2_ORF1	F4JL11	*IMPA2*	IMPORTIN SUBUNIT ALPHA-2	59.1	5	0	2	6 × 10^−61^
Sorting	151836-193_1_ORF2	P40941	*AAC2*	ADP, ATP CARRIER PROTEIN 2	42.3	7	1	5	0
Sorting	161087-178_2_ORF2	Q8H0U5	*TIC62*	PROTEIN TIC 62	73.4	6	4	6	5.77 × 10^−115^
Sorting	82340-277_1_ORF2	Q39196	*PIP1.4*	PROBABLE AQUAPORIN PIP1-4	33	6	2	3	3.75 × 10^−170^
Sorting	272341-85_2_ORF2	Q94A40	*AT1G62020*	COATOMER SUBUNIT ALPHA-1	137	1	0	1	0
Sorting	29489-466_3_ORF1	Q0WW26	*AT4G34450*	COATOMER SUBUNIT GAMMA	103	2	1	1	0
Sorting	43675-385_1_ORF2	Q67YI9	*EPSIN2*	CLATHRIN INTERACTOR EPSIN 2	85.2	1	1	1	1.51 × 10^−117^
Sorting	68824-304_5_ORF2	Q9LQ55	*DRP2B*	DYNAMIN-2B	105.3	1	1	1	0
Degradation	141778-205_4_ORF2	Q8L770	*CLPR3*	ATP-DEPENDENT CLP PROTEASE PROTEOLYTIC SUBUNIT-RELATED PROTEIN 3	38.9	2	1	1	1.73 × 10^−136^
Degradation	172993-163_5_ORF1	Q9XJ36	*CLPR2*	ATP-DEPENDENT CLP PROTEASE PROTEOLYTIC SUBUNIT-RELATED PROTEIN 2	32.7	4	1	1	4.49 × 10^−126^
Degradation	170504-166_2_ORF2	P30184	*LAP1*	LEUCINE AMINOPEPTIDASE 1	62.5	4	1	4	0

## Data Availability

The concatenated dDB is available online at http://fgcz-r-021.uzh.ch/fasta/p1222_combo_NGS_n_Viridi_20160205.fasta (accessed on 9 November 2022).

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
