# Peer review of "Proteome and Interactome Linked to Metabolism, Genetic Information Processing, and Abiotic Stress in Gametophytes of Two Woodferns"

_ijms, 2023, doi:10.3390/ijms241512429_

Round 1
Reviewer 1 Report
Authors submitted manuscript entitled “Proteome and interactome linked to metabolism, genetic information processing, and abiotic stress in gametophytes of two woodferns” to IJMS. In this study, the authors focused on ferns and lycophytes, which are less studied compared to flowering plants. They conducted proteomic analyses on the gametophytes (heart-shaped structures) of two fern species, Dryopteris affinis ssp. affinis and Dryopteris oreades, using the STRING database. They identified and analyzed a set of 218 proteins shared by these gametophytes, discussing their roles in metabolism, genetic information processing, and responses to abiotic stress. By examining the interactions among these proteins, the authors aimed to gain insights into fern function, development, and plant evolution. This manuscript needs to be revised carefully before reviewing it again.
Introduction section is not very well written. It should be expanded more by adding some relevant information.
Results:
If authors have used 218 proteins from here:
https://www.frontiersin.org/articles/10.3389/fpls.2021.718932/full#supplementary-mate-rial
Then they should make it clear in material and methods also. Authors have performed proteomic analysis in this study. Or they utilized the previous data to perform some bioinformatics analyses? Make it clear at the start of the results. If you performed the proteomic analyses from this study then mention the basic information of proteomic data (peptides, proteins, etc. and submit data to any repository)
proteomic data…. P should be capital
Authors should describe the results in detail.
The discussion section is too large and lacks critical discussion.
Authors should speculate the results and discuss them in relevance to previous data and findings from this paper. Avoid using sentences related to very basic information about some terms.
Improve Figure 5.
English is almost fine.
Author Response
Referee 1
First of all, the authors would like to let you know that your revision work is sincerely appreciated for all the authors, finding your comments/suggestions very interesting and helpful to improve the manuscript. Attached you will find the new manuscript with all the corrections done in yellow for your perusal.
Next, we are going to respond one by one, all the concerns raised by the referee.
- Introduction should be expanded more by adding some relevant information.
Many thanks for the comment. The introduction was expanded, specifying those research works published so far, using “omics” methodologies with ferns (spore, gametophyte and sporophyte), and with different scopes such as germination, reproduction, stress, etc. We think that the molecular context of ferns is now better presented.
- The referee suggests to let clear all mentioned below in material and methods, and results sections.
Thanks for the comment. In both sections a paragraph introducing the precedent work was added, so that the reader could differentiate between what we did previously, and the new reported data, which were obtained by performing bioinformatics analyses.
- Proteomic data. Put P in capital letter
Sorry, we do not know where the correction should be done.
- Authors should describe the results in detail.
Many thanks. The results were detailed as suggested by the referee. The percentage of proteins associated to several biological functions was calculated, as well as cited some of the proteins related to them. In addition, data about the list of proteins provided in tables 1 and 3, were incorporated such as the number of unique peptide count, % of coverage, total spectra, and E-value.
- The discussion is too long and lacks critical discussion. Authors should speculate the results and discuss them in relevance to previous data and findings from this paper. Avoid using sentences related to very basic information about some terms.
Thanks for the comment. In the new version, we tried to adding more data about proteins found in other fern species, to reinforce our results. Also, basic sentences about some well current known processes were removed as suggested. In other cases, some paragraphs were rewritten.
- Improve figure 5.
Many thanks for the suggestion. We have redone it, however, although the letter font size used was the same with the four pictures, the final impression after a lot of trials, is that the size letter is not exactly the same. We edited the figure by using Cytoscape, and maybe the difference relays on the different number of proteins present in each picture, but it is only speculation.
Sincerely
The authors

Reviewer 2 Report
In this manuscript, the authors utilized previously published data and performed new bioinformatic analyses to investigate the interactome of the apomictic species Dryopteris affinis and its sexual relative Dryopteris oreades.
In my opinion, this work should have been included as part of the authors' previous study and added to that paper. Therefore, the quality of this manuscript must be improved to get publish.
following are major concerns for this study:
#1. I noticed that the methods described in sections 4.1, 4.2, 4.3, and 4.4 of this study are the same as those published in the paper with the DOI: doi.org/10.3389/fpls.2021.718932. However, since you are not generating new data, what is the purpose of including these methods in this study?
#2. In your previous study (doi.org/10.3389/fpls.2021.718932), 879 proteins were quantified using the exact same protein identification method. Why is the number of identified proteins reduced to 218 in this study?
#3. Figure 3 and Figure 4 are deceptive. It seems unlikely to have such a large number of proteins associated with the ribosome and proteosome, considering the information presented in Figure 2. Additionally, the highlighted boxes, indicating shared peptides, may not necessarily imply that all the proteins have been identified. It would be beneficial to include the parameter 'unique peptide >= 2' in your protein identification step.
#4. I noticed that all the interactome analyses are based on STRING with Arabidopsis as a reference. Since these interactions are already known and are derived from different species, some verification is mendatory, but I am unable to identify any in vivo or in vitro verification steps in the manuscript.
#5. Regarding Figure 6, I would appreciate more information on the text mining parameters used. Additionally, considering that there is only one sample for each apomictic species, could you explain how the co-expression analysis was performed?
#6. The protein labels used throughout the manuscript, such as Uniprot ID, Arabidopsis ath_id, protein name, and STRING database ID, are very confusing. You should convert all accessions to protein names for consistency.
#7. Figures are in low quality, you should use same font at least.
Author Response
First of all, the authors would like to let you know that your revision work is sincerely appreciated for all the authors, finding your comments/suggestions very interesting and helpful to improve the manuscript. Attached you will find the new version of the manuscript, highlighting in yellow all the changes done for your perusal.
Next, we are going to respond one by one, all the concerns raised by the referee.
According to the referee’s suggestions, we tried to improve the quality of this manuscript as follows:
-1. I noticed that the methods described in sections 4.1, 4.2, 4.3, and 4.4 of this study are the same as those published in the paper with the DOI: doi.org/10.3389/fpls.2021.718932. However, since you are not generating new data, what is the purpose of including these methods in this study?
Thanks for the comment. Given that, we have to confess that this manuscript had been previously submitted to this journal. The revision process was so strange that we were invited to start a new round finally. The most of reviewers were very positive and we had done, as now, the modifications suggested, being one of them to explicit the methods employed in the previous proteomic analyses. In the present manuscript we have generated new data, by using bioinformatic tools from a set of 218 proteins, although the general list of proteins (879 in total) had been already published as you well mentioned, but not discussed anywhere.
-2. In your previous study (doi.org/10.3389/fpls.2021.718932), 879 proteins were quantified using the exact same protein identification method. Why is the number of identified proteins reduced to 218 in this study?
Thanks for your comment. Certainly, the number of proteins quantified initially were 879. Those referring to the differential proteins found in each fern species were reported in Fernández et al, (2021), in concrete, there were around two hundred of proteins for each species. Then, for the remaining proteins, which were shared by both species, representing around more than four hundred proteins, one half were published in the previous IJMS, and the other half are discussed in the present report.
-3. Figure 3 and Figure 4 are deceptive. It seems unlikely to have such a large number of proteins associated with the ribosome and proteosome, considering the information presented in Figure 2. Additionally, the highlighted boxes, indicating shared peptides, may not necessarily imply that all the proteins have been identified. It would be beneficial to include the parameter 'unique peptide >= 2' in your protein identification step.
Thanks for your comment. The results were provided by STRING platform. We have removed figures 3 and 4, and created the table 1, listing the proteins annotated associated to ribogenesis and proteasome. The condition “unique peptide >=2” was included in Material and methods section 4.4.
-4. I noticed that all the interactome analyses are based on STRING with Arabidopsis as a reference. Since these interactions are already known and are derived from different species, some verification is mandatory, but I am unable to identify any in vivo or in vitro verification steps in the manuscript
Thanks for your comment. The interactome uses Arabidopsis as a reference, due to the fact that most part of the protein matches came from this species. On the other hand, STRING collects information for several sources and propose the protein-protein interactions according to the deposited data. We are not aware about any sort of verification in this sense.
-5. Regarding Figure 6, I would appreciate more information on the text mining parameters used.
Thanks for your comment. According to STRING V11, automated text-mining is applied to uncover statistical and/or semantic links between proteins, based on Medline abstracts and a large collection of full-text articles. The specific information used is not provided but the scores.
Additionally, considering that there is only one sample for each apomictic species, could you explain how the co-expression analysis was performed?
We discuss here about 218 proteins shared by the gametophytes of the two selected species: D. oreades and D. affinis. The co-expression analyse is another bioinformatic tool provided by STRING platform. In brief, STRING collects gene expression evidence from a number of sources; this is then normalized, pruned, and the expression profiles over a large variety of conditions are compared. Pairs of genes that show consistent similarities between their expression profiles are assigned association scores; the majority of the expression data is RNA based, but it is also imported proteome expression data, from the Proteome HD database.
-6. The protein labels used throughout the manuscript, such as Uniprot ID, Arabidopsis ath id, protein name, and STRING database ID, are very confusing. You should convert all accessions to protein names for consistency.
Thanks for your comment. The criterion adopted is to write in the body text the protein name expanded followed by the short name (gene name) in brackets, according to Uniprot terminology. However, there are several cases, in which the only name for proteins is that starting by AT. On the other hand, the STRING platform allows us to export and edit the networks in CYTOSCAPE, and the simplest way to build the Figures 1 and 3, is that appearing in the manuscript, in which only the gene name is provided.
-7. Figures are in low quality; you should use same font at least.
Many thanks for your comment, which contributed to improved the figures. In the figure 2, we adjusted initially the font size to the number of proteins linked to each KEGG category. Regarding the new figure 3, we have to say that although the letter font size used was the same with the four pictures, the final impression reveals some subtle difference. We edited the figure by using Cytoscape, and after a lot of trials, maybe the different number of proteins in each picture could cause some influence in the final look. Finally, the figure 7 was reduced to four pictures, and the resolution was improved as much as possible, highlighting the presence of an apomictic center in D. affinis or archegonia, in D. oreades.
Sincerely
The authors

Round 2
Reviewer 1 Report
The authors have addressed my all comments.
English is almost fine, minor editing maybe required.
Reviewer 2 Report
The authors meticulously revised their original paper based on comments and suggestions. The article can now be accepted for publication in its present form.